# Resilience and its association with hopelessness, depression, loneliness, post-traumatic stress disorder (PTSD) and suicidal ideations and behavior in a cohort in the Nairobi Metropolitan

resilience; depression; hopelessness; loneliness; suicidality

**Corresponding author:**
David M. Ndetei;
Email: dmndetei@amhf.or.ke

David M. Ndetei[1,2,3] ⓘD, Victoria Mutiso[1,3], Christine Musyimi[1,3], Eric Jeremiah[1,3],

Pascalyne Nyamai[1,3], Samuel Walusaka[1,3], Veronica Onyango[1,3],

Kamaldeep Bhui[4,5] ⓘD and Daniel Mamah[6]

[1]Africa Institute of Mental and Brain Health (formerly Africa Mental Health Research and Training Foundation), Nairobi, Kenya; [2]Department of Psychiatry, University of Nairobi, Nairobi, Kenya; [3]World Psychiatric Association Collaborating Centre for Research and Training, Nairobi, Kenya; [4]Department of Psychiatry, Nuffield Department of Primary Care Health Sciences, Wadham College, University of Oxford, Oxford, UK; [5]World Psychiatric Association Collaborating Centre, Oxford, UK and [6]Department of Psychiatry, Washington University School of Medicine, St. Louis, MO, USA

## Abstract

Comprehending resilience in the face of mental health issues is important, especially for young people who deal with a variety of psychological pressures. This study aims to investigate the co-occurrence of several mental health conditions and the role of resilience as a potential intervention in youth 14–25 years in the Nairobi metropolitan area. We recruited 1,972 youths. The following self-administered instruments were used: resilience (ARM-R), hopelessness (BHS), depression (BDI, PHQ-9), PTSD (HTQ), loneliness (UCLA Loneliness Scale) and suicidality (C-SSRS). Descriptive statistics, Pearson correlation and hierarchical multiple regression analyses were conducted on the data. The key findings are that depression and hopelessness showed a strong negative association with resilience. PTSD and recent suicidal ideation and behavior showed less negative association with resilience. Building resilience is an important intervention for the conditions reported in our study among the youth. This study contributes novel insights into the intersection of multiple psychological stressors and resilience, paving the way for more targeted, integrative mental health interventions.

## Impact Statement

This study addresses a critical knowledge gap in Kenya by examining the complex interplay between resilience and co-morbid hopelessness, depression, loneliness, PTSD and suicidal ideation and behavior in the Nairobi metropolitan area. While existing research has largely focused on isolated mental health conditions in rural contexts, this study has looked at several co-morbid conditions within a culturally and socioeconomically diverse urban cohort of Kenyan youth. The research findings have significant implications for mental health interventions. The study shows that depression, hopelessness, loneliness and suicidal thoughts and behavior all have a significant and negative association with resilience. This research offers prima-facie evidence for strengthening resilience as an intervention for the studied mental health conditions in Kenyan youth living in Nairobi Metropolitan with the potential for extrapolation to similar socio-economic contexts. This study adds to the expanding body of knowledge on youth mental health in low- and middle-income urban settings and the potential role of enhancing resilience as an intervention. This study lays the foundation for controlled studies on the role of resilience in improving mental health outcomes among Kenyan youths and in similar contexts. This can be done by trained non-specialist mental health experts, including community health promoters and therefore with the potential for critical reach in the communities.





## Introduction

According to the World Health Organization (WHO) report of 2023, over 700,000 suicide cases occur worldwide annually with suicide emerging as a prominent cause of mortality among individuals aged 15–29 in 2019 (WHO, 2024), of which 77% of global suicides occur in low- and middle-income countries (LMICs). For effective and comprehensive management, there is a need to understand local context associates like resilience which may have a cumulative effect on suicidal ideas and behavior.

Depression significantly affects resilience among youths, often hindering their ability to bounce back from adversity (Mikocka-Walus et al., 2021). The negative thought patterns and emotional distress associated with depression can undermine confidence and self-efficacy, which are key components of resilience (Sabouripour et al., 2021; Trpcevska, 2017). Furthermore, depression may lead to social withdrawal and isolation, reducing access to supportive relationships that are crucial for fostering resilience (Wall, 2021).

Loneliness denies youths essential supportive relationships and resources that foster resilience (Batsleer et al., 2017; Garagiola et al., 2022). Addressing loneliness and hopelessness through interventions that promote social connections, cultivate hope and strengthen coping skills essential for nurturing resilience and supporting the well-being of youths (Betancourt et al., 2013). An increased level of loneliness is associated with decreased resilience among adolescents aged 18–25 years, impacting their ability to cope with stressors effectively (Jakobsen et al., 2020; Labrague et al., 2021; Marchini et al., 2021).

Without a sense of belonging and social support, individuals may struggle to cope effectively with life's challenges and setbacks (N. M. Lambert et al., 2013). Studies in South Africa confirm that hopelessness, characterized by a general sense of despair and doubts about the future, undermines the motivation and perseverance needed to bounce back from adversity (Chinyamurindi and Harry, 2020; Mmusi, 2020). Additionally, a study in Tanzania found that youth experiencing loneliness and hopelessness may lack belief in their ability to overcome difficulties, further diminishing their resilience when confronted with obstacles (Msangi, 2020).

Studies indicate that because of disruptions in emotional control and cognitive flexibility, those with severe PTSD symptoms may be less resilient (Yehuda et al., 2006). PTSD, especially from conflict and violence areas, has been shown to reduce resilience in Africa by making people more susceptible to distress and restricting their access to social and psychological support (Theron and Theron, 2014).

A study in Kenya found a significant correlation between resilience and a reduced risk of suicidal ideations among adolescents (Ndetei et al., 2022a). It also appears that PTSD can reduce resilience by making social disengagement, emotional dysregulation and hopelessness worse (Mbwayo et al., 2020). Another Kenyan study highlighted the prevalence of depression and its relationship with hopelessness and suicidal thoughts among university students (Ndetei et al., 2022b). Yet another study emphasized the detrimental effects of emotional negligence and loneliness on mental health outcomes in Kenyan youth (ODERA, 2021). These studies collectively underscore the vital need for comprehensive mental health interventions targeting resilience-building, depression prevention and social support mechanisms to address the growing mental health challenges in Kenya.

## Knowledge gap

Most studies in Kenya have focused on isolated cases of mental disorders rather than examining a wide range of co-occurring mental disorders within the same cohort of youth, particularly those living in metropolitan culturally diverse areas rather than purely rural settings. This study aims to fill that gap by investigating the effects of hopelessness, depression, PTSD, loneliness and suicidal ideations and behavior on resilience among youths within the Nairobi metropolitan area, encompassing diverse socioeconomic backgrounds.

## General aim

The general aim is to examine the co-occurrence of hopelessness, depression, suicidal ideation and behavior and loneliness and their relationship with resilience among youth in the Nairobi metropolitan area.

## Specific aims

1. To investigate the relationship between hopelessness and resilience among youths in the Nairobi metropolitan area.
2. To examine the association between depression and resilience.
3. To explore the relationship between resilience and PTSD.
4. To explore the relationship between resilience and loneliness at school and home.
5. To explore the relationship between resilience and suicidality.

## Methods

### Study design and setting

This study was conducted on a community sample of youth in the Nairobi Metropolitan area, not a clinical or patient population.

### Study participants

The study employed a population-based sampling approach that included youths aged 14–25 residing in the designated neighborhoods. The minimum age of 14 years reflects the entry point to high school. Exclusion criteria included individuals outside the specified age range, those unable to comprehend the questionnaire due to factors such as intoxication or illiteracy and those unwilling to participate.

## Procedure

### Research assistant selection and training

Twelve research assistants (RAs) were recruited through a competitive process designed to assess prior experiences and interpersonal skills. Subsequently, successful candidates underwent a 2-day comprehensive in-person training course, involving data collection techniques and protocols and simulated data collection role plays.

### Recruitment and resulting sample

All data was collected in the daytime from September 21, 2022 to December 15, 2022, based on a schedule that also accommodated a crucial sensitization phase through existing local networks. We approached the County Commissioners in Kiambu and Nairobi counties and upon approval linked with chiefs and sub-chiefs who worked in collaboration with elders to sensitize and mobilize the youths in their jurisdiction about the study, when and where it would be conducted and why it was important for them to participate in the study. All youth in the localities were invited to participate, provided they met the eligibility criteria, ensuring the recruitment process was as inclusive as possible.

### Data collection

The first aspect of this process entailed randomly assigning a number (1–12) to a research assistant, with each number forming a distinct group. Participants were then assigned to the group in a restricted randomization, where a printed voucher with numbers

1–12 was used to ensure group sizes were balanced. Each group comprised a maximum of 25 participants and was led by the respective RA assigned to that group. The participants were instructed to read the questions themselves.

All questionnaires were self-reported and in English, with the trained RA merely explaining the nature of the study. The RAs collected written informed consent/assent upon being signed. They also cross-checked the participants' ages to ensure compliance with the inclusion criteria. To ensure standardization, participants were instructed to respond to the questions based on their understanding, thereby minimizing any potential influence from the RA.

Data was collected on resilience, hopelessness, depression, loneliness, PTSD and suicidal ideation and behavior together with accompanying socio-demographics.

## Instruments

### Socio-demographic profile

Socio-demographic profile included age, gender, marital status, religion, birth position, level of education, employment status, primary source of income, place of abode and whether they were sharing the living space(s).

### Adult resilience measure + 16 (ARM-R)

The ARM-R is a self-report questionnaire designed to assess resilience from a socio-ecological lens (Jefferies et al., 2019; Liebenberg and Moore, 2018; Resilience Research Centre, 2022). Comprising 17 items, respondents rate each item on a five-point Likert scale, ranging from "Not at all" (1) to "A lot" (5). The ARM-R measures both personal resilience, which pertains to an individual's ability to access resources in their environment and relational resilience, which evaluates the support provided by social entities like family, peers and institutions. All items are positively framed, making scoring straightforward by summing up the scores directly. The total scores on the ARM-R range from 17 to 85, with higher scores indicating stronger resilience-related traits. ARM-R was developed through a cross-cultural study involving 14 communities across 11 countries including countries in LMICs and diverse cultures (Ungar et al., 2008). It demonstrates good psychometric properties, including strong internal reliability/consistency, content and face validity. It has been used in Syria (Ipekci, 2021), Brazil (Ferreira et al., 2022) and across cultures in diverse studies of resilience (Jefferies et al., 2019; Ungar et al., 2008). The internal consistency of the ARM-R scale was greater than 0.8 (Cronbach's alpha for 16 years and above was 0.904, and 0.898 for less than 16 years) indicating its reliability for the 14 and 15-year-olds.

### Beck Hopelessness Scale (BHS)

The Beck Hopelessness Scale (BHS) is a 20-item self-report that was originally developed by Beck and colleagues in 1974 to assess and measure the extent of negative expectations/attitudes relating to the immediate and long-term future (Beck et al., 1974). The BHS has been shown to have good internal consistency (α 0.82–0.93) and good predictive validity (0.91) (Beck et al., 1974). The Total score was computed by adding points of all the 20 items with Items 1, 3, 5, 6, 8, 10, 13, 15 and 19, FALSE = 1 point, TRUE = 0 points, and items 2, 4, 7, 9, 11, 12, 14, 16, 17, 18 and 20, TRUE = 1 point, FALSE = 0 points. The total score ranges were 0–3 = none or minimal, 4–8 = mild, 9–14 = moderate and 15–20 = severe.

### Beck's Depression Inventory scale

The Beck Depression Inventory (BDI) is a 21-item self-report instrument developed in 1961 by Beck, et al. to measure the severity of depression (Beck et al., 1961). It contains statements reflecting depressive symptoms experienced by participants in the past week, with ratings ranging from 0 to 3. Scoring is achieved by adding the highest ratings for all 21 items. The minimum score is 0 and the maximum score is 63. The total score was then categorized as follows; 1–10 = Normal, 11–16 = Mild mood disturbance, 17–20 = Borderline clinical depression, 21–30 = Moderate depression, 31–40 = Severe depression, and Over 40 = Extreme depression. The BDI has been extensively validated in various studies (Beck and Beamesderfer, 1974; Lambert et al., 1986; Snaith and Taylor, 1985). BDI takes approximately 5–10 min to complete.

### Patient Health Questionnaire-9 (PHQ-9)

The PHQ-9 is a 9-item self-report questionnaire designed to measure depression (Kroenke and Spitzer, 2002). The PHQ-9 assesses the frequency of problems respondents have experienced over the past 2 weeks. Items are rated on a four-point Likert scale, ranging from 0 (not at all) to 3 (nearly every day). The cumulative score can range from 0 to 27, with high scores meaning severe depression. The total score can then be interpreted as; minimal (0–4), mild (5–9), moderate (10–14), moderately severe (15–19), or severe (20–27). The questions address sleep, energy, appetite and other possible symptoms of depression by asking respondents how often they have "been bothered by any of the following problems" in the past two weeks. The tool has been validated in India and has good psychometric properties. It has been extensively used in many culturally diverse countries (Kochhar et al., 2007; Kroenke et al., 2001).

### The Harvard Trauma Questionnaire (HTQ)

The Harvard Trauma Questionnaire (HTQ) was developed in the early 1990s as a cross-cultural screening instrument to document trauma exposure, head trauma and trauma-related symptoms. While the original tool focused on refugee segments (Indochinese refugee populations) where it exhibited strong psychometric properties, it is increasingly being used among contexts, including low-prevalence community samples (Mollica et al., 1992). It has since been translated and validated in several languages for many regions and population samples (de Fouchier et al., 2012; Halepota and Wasif, 2001; Lhewa et al., 2007; Oruc et al., 2008). The first 16 items were derived from DSM-IV criteria for PTSD. The higher the scores on the DSM-IV PTSD items, the more likely it is that the respondent will have a PTSD diagnosis. Response options were "yes" and "no." Cumulative trauma exposure scores were calculated by a count of "yes" responses to all items.

### University of California, Los Angeles (UCLA) Loneliness scale

The UCLA Loneliness Scale is a 20-item self-report measure designed to assess an individual's subjective sense of loneliness and feelings of social isolation (Russell et al., 1978). It was used to assess participant's perceived loneliness at school and at home to see the varying impact on their mental health. Participants rate each item by selecting one of the following options: O ("I often feel this way"), S ("I sometimes feel this way"), R ("I rarely feel this way"), or N ("I never feel this way"). The scoring is calculated by assigning 3 points to all O's, 2 points to all S's, 1 point to all R's and 0 points to

all N's. The total score is obtained by summing the scores for each item together with higher scores indicating greater degrees of loneliness. The UCLA Loneliness Scale's validity and reliability has shown a high internal consistency and test–retest reliability from the college and university student's population (Russell et al., 1978; Solano, 1980).

### The Columbia–Suicide Severity Rating Scale (C-SSRS)

The Columbia–Suicide Severity Rating Scale (C-SSRS) does not introduce any new symptoms related to suicidality that have been studied widely in both HICs and LMICs. All it does is bring together all those symptoms and behaviors into single tool.

The Columbia–Suicide Severity Rating Scale (C-SSRS) is an assessment tool that assesses suicidal ideation and behavior (Posner et al., 2011). It measures four constructs to distinguish between the domains of suicidal ideation and suicidal behavior. The first is the severity of ideation (hereafter referred to as the "severity subscale"), rated on a five-point ordinal scale in which 1 = wish to be dead, 2 = non-specific active suicidal thoughts, 3 = suicidal thoughts with methods, 4 = suicidal intent and 5 = suicidal intent with plan. The second is the intensity of ideation subscale (hereafter referred to as the "intensity subscale"), which comprises five items, each rated on a five-point ordinal scale: frequency, duration, controllability, deterrents and reason for ideation. The third encompasses the behavior subscale, rated on a nominal scale that includes actual, aborted and interrupted attempts; preparatory behavior; and non-suicidal self-injurious behavior. Lastly, the fourth is the lethality subscale, which assesses actual attempts; actual lethality is rated on a six-point ordinal scale and if actual lethality is zero, potential lethality of attempts is rated on a three-point ordinal scale. Psychometric evaluations have consistently demonstrated the scale's reliability, internal consistency and construct validity, with evidence supporting its ability to accurately predict suicide risk across different populations (Austria-Corrales et al., 2023; Schwartzman et al., 2023; Yershova et al., 2016).

### Ethics

The study was approved by the Nairobi Hospital Ethics Research Committee (approval no. TNH-ERC/DMSR/ERP/022/22). The study obtained licensing from the National Commission for Science, Technology and Innovation (NACOSTI) license number NACOSTI/P/22/18097. Administrative permissions were also sought from the county-level offices in Kiambu and Nairobi counties as well as Institutional approval from the colleges. Informed written consent/assent was obtained from participants before data collection commenced. For participants younger than 18, consent to participate was obtained from their parents or legal guardians. Participants received reimbursements covering transportation, accommodation and meals. Their participation was therefore totally voluntary.

### Statistical analysis

Statistical Package for the Social Sciences (IBM SPSS Statistics) version 25 was used to perform the statistical analysis. To summarize the variable scores, descriptive statistics (mean, standard deviation (SD), range) were computed. Pearson correlation coefficients, simple linear regression and hierarchical multiple regression were used to assess the correlation and associations between resilience, hopelessness, depression, loneliness, PTSD and suicidal ideation and behavior. In hierarchical regression, blocks of variables were added to the model to evaluate the distinct contribution of each independent variable to the variance in resilience while accounting for the influence of other variables. We used both the BDI and the PHQ-9 to get a full picture of the participants' depressive symptoms. A two-way mixed-effects model with a consistency specification was used to perform an Intraclass Correlation Coefficient (ICC) study to assess the agreement between PHQ-9 and BDI to determine if the 2 measures offered complementing information on depression symptoms or were interchangeable. Collinearity diagnostics were performed to assess potential redundancy between the independent variables. The variance inflation factor (VIF) values for all independent variables were below 5 (1.163 to 3.33), indicating that multicollinearity was not a concern in our model. The significance level for all analyses was set at $p < 0.05$.

## Results

### Social demographics

The study had a total of 1,972 participants aged 14–25 years ($M = 20.49$, SD = 2.631), with more than half females (55%). Regarding religion, 87.2% were Christians while 5.5% were Muslim. Most participants were at the secondary level (52.4%) with the least being in university (6.9%). The majority were unmarried (59.9%) and unemployed (77.6%) as they were mostly students. The majority rented a house (62.6%) and resided in urban areas (91.5%).

### Frequency of the resilience items

The response rates to the different items of ARM-R were high (98–99%). The lowest scores (not at all) ranged from 5–15%. These lowest scores were accounted for by "If I am hungry, I get food to eat" (14.3%), "I talk to my family/partner about how I feel (14.9%) and "My family has usually supported me throughout life" (5.3%). Overall, well over 50% scored positively in combined "quite a lot" and "a lot."

### Descriptive statistics for various psychosocial variables used

Overall resilience as measured by a composite score shows a mean of 64.36 (SD = 14.374) with a range from 17 to 85. Mental health indicators include BHS total score (mean = 3.96, SD = 3.353), BDI total score (mean = 14.68, SD = 11.864) and PHQ score (mean = 7.45, SD = 5.576). Loneliness scores for school and home environments are 23.71 (SD = 12.509) and 22.73 (SD = 12.489), respectively. Participants report low levels of suicidal ideation with a mean lifetime score of 0.9708 (SD = 1.49633) and a recent score of 0.5695 (SD = 1.18114). The PTSD score averages 6.44 (SD = 2.559). See Table 1

### The correlation between resilience, hopelessness, depression, PTSD, loneliness and suicidal ideation and behavior

From Table 2, resilience exhibited moderate negative relationship with psychological measures, including the PHQ score ($r = -0.361$, $p < 0.01$), BDI total score ($r = -0.465$, $p < 0.01$), BHS total score ($r = -0.446$, $p < 0.01$), loneliness scores at school ($r = -0.408$, $p < 0.01$) and home ($r = -0.391$, $p < 0.01$), as well as low negative correlation with PTSD, suicide ideation and suicide behavior.

**Table 1.** Variable scores and mean (SD; range)

| Variables | Mean (SD; range) |
| --- | --- |
| Overall resilience | 64.36(14.374;17–85) |
| Personal resilience | 37.07(8.952;10–50) |
| Relational resilience | 26.92(6.459;7–35) |
| BHS total score | 3.96(3.353;0–19) |
| BDI total score | 14.68(11.864;0–64) |
| PHQ score | 7.45(5.576;0–27) |
| Loneliness score school | 23.71(12.509;0–60) |
| Loneliness score home | 22.73(12.489;0–60) |
| PTSD score | 6.44(2.559;0–11) |
| Suicide ideation scores lifetime | 0.97(1.50; 0–5) |
| Suicide ideation scores recent | 0.57(1.18;0–5) |
| Suicide behavior lifetime | 0.62(1.82; 0–11) |
| Suicide behavior recent | 0.29 (1.21; 0–11) |

SD, standard deviation.

### Regression analysis

When comparing the PHQ-9 and BDI, the ICC analysis showed a moderate level of agreement (single measures ICC = 0.429, 95% CI: 0.391–0.465; average measures ICC = 0.600, 95% CI: 0.562–0.635), indicating that although the measures are connected, they evaluate different but overlapping features of depression.

Lower levels of resilience were substantially associated with greater levels of hopelessness, depression, PTSD symptoms, suicidal ideation and behavior, and loneliness at home and school (all $p < 0.001$). Depression had the highest effect on resilience, with each unit increase in the standard deviation in depression scores associated with a decrease in resilience of 0.465 units ($p < 0.001$), followed by hopelessness, where each standard unit increase in hopelessness scores was associated with a decrease in resilience of 0.446 units ($p < 0.001$). PTSD symptoms had a slightly lower impact, with each standard deviation unit increase associated with a decrease in resilience of 0.141 units ($p < 0.001$). See Table 3

### Hierarchical analysis of resilience and hopelessness, depression, loneliness, PTSD and suicidality (ideation and behavior)

From Table 4, the *R*-squared values indicate the proportion of variability in resilience explained by the models. Models 1–4 show progressively increasing *R*-squared values (0.178–0.3), suggesting an improved ability to account for the variance in resilience as additional variables are included. The adjusted *R*-square values, which account for the number of independent variables in the model, also show a similar pattern for models 1–5, ranging from 0.172 to 0.246 except for model 3 which shows a decrease when PTSD is added implying its low account in associating with resilience. Hopelessness consistently exhibits a strong negative association with resilience, with standardized coefficients ranging from −0.422 to −0.259 across different models, emphasizing its role in impacting resilience. Lifetime suicidal behavior also contributes to the model significantly showing its impact on resilience. PTSD, loneliness, suicidal ideation and recent suicidal behavior were non-significant when introduced showing their low account in impacting resilience.

## Discussion

### Preamble

We present the first study in Kenya that investigates the associations of suicidal ideas and behavior, (the usual starting points towards suicide), and co-morbid resilience with several mental health indicators studied in the same cohort of participants in a metropolitan setting. We used a wide variety of globally used tools with good psychometric properties. We used different statistical methods, all of which generated the same conclusion. Our findings suggest potential interventions and policy formulations for youths in Nairobi Metropolitan to enhance inclusion in suicidal ideas and behaviors.

### Descriptive statistics for various psychosocial variables among the participants

Our findings underscore the significance of resilience in mitigating mental health challenges. The overall high resilience mean score was 64.36, while personal resilience and relational resilience had low mean scores of 37.07 and 26.92, respectively. These suggest diverse coping mechanisms, with variable levels of personal and relational resilience affecting mental health outcomes among youths. This indicates that personal factors, such as self-efficacy and problem-solving skills contribute significantly to resilience among students as found by (Şenocak and Demirkıran, 2023) while relational resilience contributed to extreme loneliness in both school (*M* = 23.71) and home (*M* = 22.73) environments, reflecting the impact of social support on well-being. Our findings agree with those of (Nowicki, 2008) that students who receive support from home achieve significantly better grades than those who receive less support. The mean depression scores for the BDI and PHQ-9 fall within mild depression. The mild depressive symptoms of our community sample rather than a clinical one could indicate underlying psychosocial challenges. Further study is needed to explore the broader mental health context in this group to inform early interventions and prevention strategies. While participants reported low levels of suicidal ideation (*Lifetime: M = 0.9708; Recent: M = 0.5695*) and moderate PTSD symptoms (*M = 6.44*), the relationship between resilience and these outcomes calls for further investigation.

### The associations between resilience, hopelessness, depression, loneliness, PTSD and suicidal ideation and behavior

Overall, our study underscores the relationships among psychosocial factors, highlighting the pivotal role of resilience in mental health outcomes. Our findings of a strong positive correlation between overall resilience and both personal (*r* = 0.956) and relational resilience (*r* = 0.913), indicate a cohesive relationship between different aspects of resilience. Moreover, depression, assessed by the PHQ-9 and BDI, demonstrates negative correlations with overall resilience, personal resilience and relational resilience, suggesting that higher depression levels are associated with low resilience levels, similar to other studies (Silk et al., 2007; Zhang et al., 2020).

Loneliness scores, either at home or school, exhibit moderate negative correlations with overall resilience, highlighting the adverse impact of loneliness on overall resilience levels. For example, when an individual experiences depression, as shown by a higher PHQ-9 Score, there is an increase in loneliness at school,

**Table 2.** The associations between resilience, hopelessness, depression, loneliness, PTSD and suicidal ideation and behavior

| Measures | Overall resilience | Personal resilience | Relational resilience | PHQ Score | BDI Total score | BHS Total Score | Loneliness Score School | Loneliness Score Home | PTSD Score | Suicide ideation scores Lifetime | suicide ideation scores Recent | Suicide Behavior Lifetime | Suicide behavior recent |
|---|---|---|---|---|---|---|---|---|---|---|---|---|---|
| Overall resilience | 1 | | | | | | | | | | | | |
| Personal resilience | .956** | 1 | | | | | | | | | | | |
| Relational resilience | .913** | .754** | 1 | | | | | | | | | | |
| PHQ score | −.361** | −.326** | −.341** | 1 | | | | | | | | | |
| BDI total score | −.465** | −.452** | −.420** | .554** | 1 | | | | | | | | |
| BHS total Score | −.446** | −.432** | −.400** | .348** | .496** | 1 | | | | | | | |
| Loneliness score School | −.408** | −.423** | −.323** | .422** | .367** | .276** | 1 | | | | | | |
| Loneliness score home | −.391** | −.358** | −.370** | .441** | .421** | .327** | .671** | 1 | | | | | |
| PTSD score | −.141** | −.107** | −.141** | .332** | .163** | .062* | .209** | .241** | 1 | | | | |
| Suicide ideation scores lifetime | −.205** | −.181** | −.182** | .341** | .283** | .209** | .186** | .318** | .144** | 1 | | | |
| suicide ideation scores recent | −.192** | −.172** | −.172** | .297** | .274** | .265** | .191** | .237** | .110* | .602** | 1 | | |
| Suicide behavior lifetime | −.236** | −.208** | −.221** | .253** | .243** | .159** | .161** | .215** | .134** | .561** | .352** | 1 | |
| Suicide behavior recent | −.201** | −.204** | −.180** | .175** | .181** | .185** | .203** | .189** | .102** | .458** | .514** | .736** | 1 |

*Correlation is significant at the 0.05 level (2-tailed).
**Correlation is significant at the 0.01 level (2-tailed).
Cohen's guidelines for interpretation: small (0.10 to 0.29), medium (0.30 to 0.49) and high (0.50 to 1.00).

**Table 3.** Simple linear regression analysis of resilience and hopelessness, depression, PTSD, loneliness (home and school) and suicidality (ideation and behavior)

| Model | Variables | Unstandardized coefficients | | Standardized coefficients | | |
|---|---|---|---|---|---|---|
| | | B | Std. error | Beta (β) | t-Value | Sig. |
| 1 | BHS total score | −1.893 | 0.098 | −0.446 | −19.346 | p < 0.001 |
| 2 | BDI total score | −0.586 | 0.027 | −0.465 | −21.798 | p < 0.001 |
| 3 | PHQ–9 score | −0.914 | 0.058 | −0.361 | −15.743 | p < 0.001 |
| 4 | PTSD | −0.765 | 0.146 | −0.141 | −5.226 | p < 0.001 |
| 5 | Suicide ideation lifetime | −2.055 | 0.372 | −0.205 | −5.526 | p < 0.001 |
| 6 | Suicide ideation recent scores | −2.465 | 0.516 | −0.192 | −4.779 | p < 0.001 |
| 7 | Suicide behavior lifetime | −1.766 | 0.214 | −0.236 | −8.238 | p < 0.001 |
| 8 | Suicide behavior recent | −2.259 | 0.355 | −0.201 | −6.368 | p < 0.001 |
| 9 | Loneliness at school | −0.44 | 0.032 | −0.408 | −13.588 | p < 0.001 |
| 10 | Loneliness at home | −0.44 | 0.027 | −0.391 | −16.218 | p < 0.001 |

BHS, Beck's Hopelessness Scale; BDI, Beck's Depression Inventory Scale; PTSD, post-traumatic stress disorder; PHQ-9, Patient Health Questionnaire-9. Models 1–10; simple linear regression; dependent-resilience.

signifying the potential rise in loneliness in the context of depressive symptoms. This aligns with findings that higher loneliness scores are associated with higher levels of depressive symptoms among college students (Qirtas et al., 2023).

The finding that suicidal ideation scores in both lifetime and recent are negatively correlated with overall resilience suggests that the higher the resilience the lower the suicidal ideations. This underscores the protective role of resilience against suicidal thoughts. This is in agreement with the findings of the longitudinal study done by Chen and Kuo (2020), examining the effects of perceived stress and resilience on suicidal ideation among early adolescents, where they found that problem-solving and perceptive maturity aspects of resilience showed a significant protective effect on suicidal ideations.

### Simple Linear regression analysis of resilience and hopelessness, depression, PTSD, loneliness (home and school) and suicidality (ideation and behavior)

Our findings also emphasize the significant effects of different psychological factors on resilience. Depression emerged as the most significant factor, with each standard deviation increase associated with a decrease in resilience, followed by hopelessness. Compared to other studies, resilience can buffer against PTSD (Rakesh et al., 2019), while recent research also indicates that a mix of environmental and genetic variables may contribute to resilience and PTSD (Thompson et al., 2018; Wolf et al., 2018). Resilience can be an outcome and also a moderating factor that buffers against the negative psychological impact of trauma (Fino et al., 2020). This highlights the complexity of the relationship between PTSD and resilience, suggesting that interventions should address both genetic predispositions and modifiable environmental factors to effectively enhance resilience and reduce the impact of PTSD.

Moreover, this study identified significant associations between resilience and suicidality, including suicidal ideation and suicidal behavior, underscoring the intricate relationship between resilience and risk factors for suicide.

A previous study that examined factors that influence the resilience of depressed young people to suicidal ideation and suicide attempts found that there is a significance between the two variables (Fergusson et al., 2003). Loneliness both at home and at school, also significantly impacted resilience, emphasizing the importance of social support in fostering resilience.

### Hierarchical analysis of resilience and hopelessness, depression, loneliness, PTSD and suicidality (ideation and behavior)

The hierarchical analysis of resilience and various psychological factors; hopelessness, PTSD, loneliness, depression and suicidality emphasizes the relationship between these psychological factors and their effect on resilience among the youth. The R-squared values showed the variance in resilience explained by the models. We found that Model 5 had the highest R-squared value of 0.3, implying an improved ability to account for resilience variance as additional variables are included. In addition, hopelessness consistently had a strong negative association with resilience across all models, highlighting its significant role in impacting resilience. This aligns with previous research that has identified hopelessness as a key predictor of decreased resilience (Satici, 2016). Suicide behavior lifetime also significantly contributes to the model, emphasizing its negative effect on resilience, corroborating findings by Chen and Kuo (2020) and Siegmann et al. (2018) who found that there is a strong association between suicidal behavior and decreased resilience.

To ensure a robust measurement of depressive symptoms, the model used both PHQ-9 and BDI. Concordance analysis revealed moderate agreement between the two measurements, with an ICC of 0.429 for single measures and 0.600 for average measures. This moderate agreement suggests that while the PHQ-9 and BDI measure overlapping aspects of depression, each contributes unique information, justifying their use together in regression models.

What was more noteworthy is that PTSD, suicide ideation, loneliness and recent suicide behavior show non-significance when introduced into the models, indicating their low effect on resilience. This contrasts with a previous study that found significant relationships between PTSD symptoms and resilience (Thompson et al., 2018), suggesting potential nuances in the relationship between these variables across different populations or contexts. While hopelessness and suicide behavior emerge as significant associations with resilience, the non-significance of PTSD, suicidal ideation and loneliness calls for further investigation to better understand their role in resilience among young adolescents.

**Table 4.** Hierarchical analysis of resilience and hopelessness, depression, PTSD, loneliness and suicidality (ideation and behavior)

| Model | R-square | Adjusted R-square | Variables | Unstandardized coefficients | Std. error | Standardized coefficients | t | Sig. |
|---|---|---|---|---|---|---|---|---|
| | $R^2$ | $R^2$ adj | | B | SE | Beta (β) | | |
| 1 | 0.178 | 0.172 | BHS total score | −1.357 | 0.248 | −0.422 | −5.469 | ***p < 0.001*** |
| 2 | 0.243 | 0.227 | BHS total score | −0.838 | 0.286 | −0.261 | −2.933 | **0.004** |
| | | | BDI total score | −0.226 | 0.140 | −0.186 | −1.621 | 0.107 |
| | | | PHQ–9 score | −0.299 | 0.231 | −0.140 | −1.294 | 0.198 |
| 3 | 0.262 | 0.235 | BHS total score | −0.807 | 0.285 | −0.251 | −2.836 | **0.005** |
| | | | BDI total score | −0.176 | 0.142 | −0.145 | −1.238 | 0.218 |
| | | | PHQ–9 score | −0.187 | 0.239 | −0.087 | −0.784 | 0.434 |
| | | | Loneliness at school | −0.011 | 0.094 | −0.011 | −0.119 | 0.906 |
| | | | Loneliness at home | −0.157 | 0.106 | −0.158 | −1.484 | 0.140 |
| 4 | 0.262 | 0.229 | BHS total score | −0.814 | 0.287 | −0.253 | −2.832 | **0.005** |
| | | | BDI total score | −0.175 | 0.143 | −0.144 | −1.226 | 0.222 |
| | | | PHQ–9 score | −0.18 | 0.242 | −0.084 | −0.745 | 0.458 |
| | | | Loneliness at school | −0.012 | 0.094 | −0.012 | −0.123 | 0.902 |
| | | | Loneliness at home | −0.153 | 0.108 | −0.154 | −1.425 | 0.157 |
| | | | PTSD | −0.086 | 0.415 | −0.016 | −0.208 | 0.836 |
| 5 | 0.300 | 0.246 | BHS total score | −0.833 | 0.293 | −0.259 | −2.843 | **0.005** |
| | | | BDI total score | −0.193 | 0.145 | −0.158 | −1.326 | 0.187 |
| | | | PHQ–9 score | −0.06 | 0.252 | −0.028 | −0.238 | 0.812 |
| | | | Loneliness at school | −0.069 | 0.103 | −0.07 | −0.665 | 0.507 |
| | | | Loneliness at home | −0.116 | 0.111 | −0.116 | −1.036 | 0.302 |
| | | | PTSD | −0.013 | 0.415 | −0.002 | −0.031 | 0.975 |
| | | | Suicide ideation lifetime | 0.854 | 1.064 | 0.097 | 0.803 | 0.423 |
| | | | Suicide ideation recent scores | −0.396 | 1.220 | −0.035 | −0.324 | 0.746 |
| | | | Suicide behavior lifetime | −1.963 | 0.809 | −0.326 | −2.427 | **0.017** |
| | | | Suicide behavior recent | 1.150 | 1.048 | 0.148 | 1.098 | 0.274 |

*Note*: Dependent variable: Resilience. BHS, Beck's Hopelessness Scale; BDI, Beck's Depression Inventory Scale; PTSD, post-traumatic stress disorder; PHQ-9, Patient Health Questionnaire-9. Bolded *p*-values are significant. *R*-squared ($R^2$): A statistical measure that represents the proportion of the variance in the dependent variable that is predictable from the independent variables. An *R*-squared value closer to 1 indicates a better fit of the model to the data. Adjusted *R*-squared: A modified version of *R*-squared that adjusts for the number of predictors in the model, provides a more accurate measure of model fit, especially when comparing models with different numbers of predictors. Unlike *R*-squared, adjusted *R*-squared can decrease if unnecessary predictors are included in the model.

The final regression model's adjusted *R*-squared value of 0.246 indicates that a large amount of resilience variance is still unaccounted for, although statistically significant. This suggests that there are probably more factors affecting resilience scores than those included in the current investigation.

Future research could benefit from examining other predictors of resilience, such as environmental, familial and individual characteristics, as well as possible mediators and moderators that might offer a broader perspective of the variables influencing resilience.

Evidence-based approaches like cognitive-behavioral therapy (CBT) can be useful in treating depression, hopelessness and PTSD (Antipas, 2022; WANGILA, 2023). Furthermore, encouraging social–emotional learning (SEL) programs in schools can assist young people in developing resilience by imparting coping mechanisms, emotional control and interpersonal skills all of which are essential for stress management and averting mental health emergencies (Norman et al., 2022). Integrating resilience-building initiatives into mental health frameworks is crucial from a policy perspective. Policies that focus on setting up youth-centered mental health promotion programs that are cost-effective, affordable, available and evidence-based in community centers and schools that offer both therapeutic and preventative assistance are to be encouraged. These policies include incorporating mental health education programs into school curricula on coping mechanisms, promotion of help-seeking behaviors and de-stigmatization of mental health disorders (Robinson, 2024). This calls for a collaborative approach between the government, local communities and healthcare providers.

## Limitations and recommendations

While efforts were made to include a diverse range of participants, the sample is limited to those who responded to community sensitization and those already in educational institutions.

Our study design is cross-sectional, which limits our ability to establish causality between resilience and mental health outcomes.

With an adjusted *R*-squared value of 0.246, a sizable amount of the variability in resilience cannot be explained. Future studies should consider other variables that may influence resilience, such as socioeconomic status, coping strategies and social support. In addition, the impacts of mediation and moderation could be investigated with coping mechanisms or social support as potential mediators or moderators.

## Conclusion

In conclusion, our study provides valuable insights into the relationship between psychological factors and resilience among youths in the Nairobi metropolitan area. We found some significant associations between hopelessness, depression, suicidal ideation, loneliness and resilience, highlighting the critical role of resilience in mitigating mental health challenges. This calls for strategies to strengthen resilience. Depression was the most influential factor, which was followed by hopelessness, with both showing substantial negative impacts on resilience.

Furthermore, our study identified significant associations between resilience and suicidality, emphasizing the protective role of resilience against suicidal thoughts and behaviors. However, some factors like PTSD, suicidal ideation and loneliness showed no significance in certain models, suggesting potential future studies in exploring their relationship with resilience among the youth population.

**Open peer review.** To view the open peer review materials for this article, please visit http://doi.org/10.1017/gmh.2025.27.

**Data availability statement.** Requests for the data may be sent to the corresponding author.

**Author contributions.** D.M.N. – conceptualization, drafting of the paper; D.W. – critique of the manuscript; V.M. – oversight of data collection; K.B. – critique of the manuscript; J.R.S. – critique of the manuscript; C.M. – oversight on ethics; S.C.W. – critique of the manuscript; P.N. – draft review; S.W. – statistical analysis; V.O. – field work during data collection and literature review; E.J. – statistical analysis and literature review; T.L.O. – critique of the manuscript; D.M. – conceptualization and critique of the manuscript. All authors read and approved the final manuscript.

**Financial support.** This study was funded by the National Institutes of Health (NIH), Grant/Award number: 5R01MH127571–02.

**Competing interest.** The authors declare no competing interests.

**Ethics statement.** The study was approved by the Nairobi Hospital Ethics Research Committee (approval no. TNH-ERC/DMSR/ERP/022/22). The study obtained licensing from the National Commission for Science, Technology and Innovation (NACOSTI) license number NACOSTI/P/22/18097. Administrative permissions were also sought from the county-level offices in Kiambu and Nairobi counties, along with Institutional approval from the colleges. Informed written consent/assent was obtained from participants before data collection commenced. For participants under the age of 18, consent to participate was obtained from their parents or legal guardians. Participants received reimbursements covering transportation, accommodation and meals. Their participation was therefore totally voluntary.

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
