## [Reviewer Report]

This is an interesting study, on a large sample of youths in urban-semiurban settings in Kenya, focusing on the association among several psychosocial variables and the construct of resilience. It found several associations, with hopelessness and depression emerging as important ‘predictors’ of resilience on hierarchical regression analysis. The findings have some important implications for youth mental health at the level of public mental health, though that is beyond the scope of this study.

One important limitation of the manuscript is the complete lack of any mention of limitations! Certainly no study is entirely free of any limitation. Some of the relevant limitations that need to be acknowledged and addressed in the discussion section include (but are not limited to):

1. This is an ‘association’ study, which does not show causality in any direction. Although the regression analysis was conducted with resilience as the “dependent” variable, regression analysis is not really a prediction study. Replacing resilience with hopelessness as the “dependent” variable, for example, would have shown, perhaps equally strongly, that (lack of) resilience is a predictor of hopelessness, not the other way round.

2. In the same line, this was a cross-sectional study, with all the variables studied concomitantly. Almost all the variables studied are dynamic over time. It would have been important to conduct at least a two-point data collection to capture the dynamic process and relations among the study variables better, and could have come a step closer to using a predictive model.

3. It is not clear if all the study instruments were available in a valid shape for use as self-administered questionnaires in local language and was comprehensible by the study subjects. This is important because all the data are based on self report.

4. An adjusted R squared value of 0.246 in the final model of the regression, while not negligible in social sciences, is not a robust value either, indicating that there would be other important variables contributing to the resilience scores in the study sample.

5. The mean values of BDI and especially PHQ-9 fall in the range of mild depression. Given that this was a community sample and not a patient population (though not made very clear in the methodology), would the authors like to comment upon that?

6. In this regard, it would be important to know how such an impressively large sample was collected. What was the method? Was there any incentives for participation in the study? Was any attempt made at making the sample representative of the population or was it entirely a convenience sample? If so, all these must be addressed and mentioned as limitations.

Other than these, a few minor issues need addressing.

1. Page 5, General aim - “The study aims to examine the co-morbid of hopelessness, depression, suicidal ideations...” this sentence needs clarity and paraphrasing.

2. The title misses the focus of this paper on resilience, which becomes apparent from the aim statement onward. The authors may want to suitably moderate the title to reflect this focus.

3. The results show loneliness data as “At school” and “At home” whereas that is not mentioned while describing the scale under methodology.

4. Page 14, first paragraph: there are many double asterix signs, which probably indicates statistical significance but are confusing; these should be removed and P values mentioned instead.

5. Some of the references are incomplete and inaccessible; please rectify, e.g.,

Batsleer, J., Duggan, J., McNicol, S., Spray, S., & Angell, K. (2017). Loneliness Connects Us.42nd Street..

Centre, R. R. (2018). CYRM and ARM user manual. Resilience Research Centre.

Ipekci, B. (2021). Posttraumatic Stress and Resilience Among Iraqi and Syrian Refugees. University of Massachusetts Boston.

Labrague, L. J., JAAD, S., & Falguera, C. (2021). Social and emotional loneliness among college students during the COVID-19 pandemic..... (publication details missing)

Nowicki, A. (2008). Self-efficacy, sense of belonging and social support as predictors of

resilience in adolescents. (Publication details missing)

ODERA, O. W. (2022). ATTACHMENT STYLES AS PREDICTORS OF BEHAVIORS AMONG SECONDARY SCHOOL STUDENTS IN NAIROBI COUNTY, KENYA. (Publication details missing)

WHO. (2023). Suicide - key facts (publication and access details missing)

---

## [Reviewer Report]

1. The abstract should have a clearer statement of the study’s novelty or unique contribution to the field.

2. The rationale for including participants aged 14–25 could be clarified. Why was this age range chosen, and does it align with resilience and mental health development studies?

3. The exclusion criteria (e.g., illiteracy and intoxication) need justification. How this might affect the generalizability of the findings?

4. The decision to use multiple tools for depression (BDI and PHQ-9) without explaining how discrepancies were handled could lead to confusion. There should be discussion arount this aspect.

5. The paper does not mention whether informed consent was obtained from participants

6. The paper does not discuss multicollinearity or other assumptions of regression analysis, which might impact the validity of the findings. For example, depression and hopelessness are likely correlated, which could affect the regression results. This should be addressed properly.

7. Tables in results section are not self-explanatory. For instance, the meaning of certain terms (e.g., “Adjusted R-Squared”) might not be clear to all readers without further elaboration.

8. The interpretation of correlation coefficients (e.g., “moderate negative associations”) should align with established thresholds (e.g., Cohen’s guidelines).

9. The discussion on the protective role of resilience needs more nuanced insights. For example, why is PTSD less impactful on resilience than expected, and how does this finding compare with other studies?

10. Discussion should also include on how the findings can translate into tangible mental health strategies or policies.

---

## [Editor Report]

Dear Author,

I hope this email finds you well.

Thank you for submitting your manuscript, “The association between resilience, hopelessness, depression, loneliness, post-traumatic stress disorder (PTSD) and suicidal ideations and behaviour in a cohort in the Nairobi Metropolitan”, to Cambridge Prisms: Global Mental Health. The reviewers have provided thoughtful and constructive feedback on your manuscript, and I believe their suggestions will be valuable in strengthening the overall quality and clarity of your work.

I would like to ask you to carefully review the comments provided by the reviewers and consider their suggestions in your revision. In particular, please ensure that you respond to each of the reviewers' points either by implementing their suggestions or providing a clear rationale for why you may choose not to incorporate certain recommendations.

Below, I’ve listed the specific comments from each reviewer for your reference:

Reviewer 1: 

This is an interesting study, on a large sample of youths in urban-semiurban settings in Kenya, focusing on the association among several psychosocial variables and the construct of resilience. It found several associations, with hopelessness and depression emerging as important ‘predictors’ of resilience on hierarchical regression analysis. The findings have some important implications for youth mental health at the level of public mental health, though that is beyond the scope of this study.

One important limitation of the manuscript is the complete lack of any mention of limitations! Certainly no study is entirely free of any limitation. Some of the relevant limitations that need to be acknowledged and addressed in the discussion section include (but are not limited to):

1. This is an ‘association’ study, which does not show causality in any direction. Although the regression analysis was conducted with resilience as the “dependent” variable, regression analysis is not really a prediction study. Replacing resilience with hopelessness as the “dependent” variable, for example, would have shown, perhaps equally strongly, that (lack of) resilience is a predictor of hopelessness, not the other way round.

2. In the same line, this was a cross-sectional study, with all the variables studied concomitantly. Almost all the variables studied are dynamic over time. It would have been important to conduct at least a two-point data collection to capture the dynamic process and relations among the study variables better, and could have come a step closer to using a predictive model.

3. It is not clear if all the study instruments were available in a valid shape for use as self-administered questionnaires in local language and was comprehensible by the study subjects. This is important because all the data are based on self report.

4. An adjusted R squared value of 0.246 in the final model of the regression, while not negligible in social sciences, is not a robust value either, indicating that there would be other important variables contributing to the resilience scores in the study sample.

5. The mean values of BDI and especially PHQ-9 fall in the range of mild depression. Given that this was a community sample and not a patient population (though not made very clear in the methodology), would the authors like to comment upon that?

6. In this regard, it would be important to know how such an impressively large sample was collected. What was the method? Was there any incentives for participation in the study? Was any attempt made at making the sample representative of the population or was it entirely a convenience sample? If so, all these must be addressed and mentioned as limitations.

Other than these, a few minor issues need addressing.

1. Page 5, General aim - “The study aims to examine the co-morbid of hopelessness, depression, suicidal ideations...” this sentence needs clarity and paraphrasing.

2. The title misses the focus of this paper on resilience, which becomes apparent from the aim statement onward. The authors may want to suitably moderate the title to reflect this focus.

3. The results show loneliness data as “At school” and “At home” whereas that is not mentioned while describing the scale under methodology.

4. Page 14, first paragraph: there are many double asterix signs, which probably indicates statistical significance but are confusing; these should be removed and P values mentioned instead.

5. Some of the references are incomplete and inaccessible; please rectify, e.g.,

Batsleer, J., Duggan, J., McNicol, S., Spray, S., & Angell, K. (2017). Loneliness Connects Us.42nd Street..

Centre, R. R. (2018). CYRM and ARM user manual. Resilience Research Centre.

Ipekci, B. (2021). Posttraumatic Stress and Resilience Among Iraqi and Syrian Refugees. University of Massachusetts Boston.

Labrague, L. J., JAAD, S., & Falguera, C. (2021). Social and emotional loneliness among college students during the COVID-19 pandemic..... (publication details missing)

Nowicki, A. (2008). Self-efficacy, sense of belonging and social support as predictors of

resilience in adolescents. (Publication details missing)

ODERA, O. W. (2022). ATTACHMENT STYLES AS PREDICTORS OF BEHAVIORS AMONG SECONDARY SCHOOL STUDENTS IN NAIROBI COUNTY, KENYA. (Publication details missing)

WHO. (2023). Suicide - key facts (publication and access details missing)

Reviewer 2:

1. The abstract should have a clearer statement of the study’s novelty or unique contribution to the field.

2. The rationale for including participants aged 14–25 could be clarified. Why was this age range chosen, and does it align with resilience and mental health development studies?

3. The exclusion criteria (e.g., illiteracy and intoxication) need justification. How this might affect the generalizability of the findings?

4. The decision to use multiple tools for depression (BDI and PHQ-9) without explaining how discrepancies were handled could lead to confusion. There should be discussion arount this aspect.

5. The paper does not mention whether informed consent was obtained from participants

6. The paper does not discuss multicollinearity or other assumptions of regression analysis, which might impact the validity of the findings. For example, depression and hopelessness are likely correlated, which could affect the regression results. This should be addressed properly.

7. Tables in results section are not self-explanatory. For instance, the meaning of certain terms (e.g., “Adjusted R-Squared”) might not be clear to all readers without further elaboration.

8. The interpretation of correlation coefficients (e.g., “moderate negative associations”) should align with established thresholds (e.g., Cohen’s guidelines).

9. The discussion on the protective role of resilience needs more nuanced insights. For example, why is PTSD less impactful on resilience than expected, and how does this finding compare with other studies?

10. Discussion should also include on how the findings can translate into tangible mental health strategies or policies.

We look forward to receiving your revised manuscript and response to the reviewers’ feedback. If you have any questions or need further clarification, please don’t hesitate to reach out.

Best regards, Sara Romero